

# Determination of drought tolerance of different strawberry genotypes

Eser Celiktopuz[1,2]

[1] Biological and Agricultural Engineering, Texas A&M University, College Station, TX, United States
[2] Agricultural Structures and Irrigation, Cukurova University, Adana, Turkey

Corresponding author
Eser Celiktopuz,
eceliktopuz@gmail.com

## ABSTRACT

Strawberry production future depends on productive, high quality and drought tolerant varieties. The goal of this study was to determine the most suitable variety by determining the yield and photosynthetic responses (net photosynthesis (Pn), stomatal conductance ($g_s$), and transpiration rate (E)) of four strawberry genotypes with different characteristics (Rubygem, Festival; 33, and 59) at two different irrigation levels (IR50: water stress (WS), IR100: well-watered (WW)). It was also aimed to prepare the irrigation program by making use of the crop water stress index (CWSI). The trial was conducted at the Agronomic Research Area, University of Çukurova, Turkey during 2019–2020 experimental year. The trial was implemented as a 4 × 2 factorial scheme of genotypes and irrigation levels, in a split-plot design. Genotype Rubygem had the highest canopy temperature (Tc)–air temperature (Ta), whereas genotype 59 had the lowest, indicating that genotype 59 has better ability to thermoregulate leaf temperatures. Moreover, yield, Pn, and E were found to have a substantial negative relationship with Tc–Ta. WS reduced yield, Pn, $g_s$, and E by 36%, 37%, 39%, and 43%, respectively, whereas it increased CWSI (22%) and irrigation water use efficiency (IWUE) (6%). Besides, the optimal time to measure leaf surface temperature of strawberries is around 1:00 pm and strawberry irrigation management might be maintained under the high tunnel in Mediterranean utilizing CWSI values between 0.49 and 0.63. Although genotypes had varying drought tolerance, the genotype 59 had the strongest yield and photosynthetic performances under both WW and WS conditions. Furthermore, 59 had highest IWUE and lowest CWSI in the WS conditions, proving to be the most drought tolerant genotype in this research.

## INTRODUCTION

The most fundamental problem of the 21st century worldwide is to supply sustainable food in sufficient quantities for the ever-increasing human population. The Food and Agriculture Organization of the United Nations (FAO) predicts that by 2050, the human population will reach 9.4–10.1 billion and the food supply based on agriculture will increase in this direction (*United Nations, 2019*). In addition, the severity of the effects such as global warming due to climate change, decrease in the number of production areas and pressures on water resources tend to increase day by day. In this context, the future of

strawberry production depends on productive, high quality and drought tolerant varieties. However, strawberry genotypes respond differently to drought stress (*Giné-Bordonaba & Terry, 2016*). *Fragaria chiloensis* was shown to be more drought resistant than *F. virginiana* and *Fragaria × ananassa* among strawberry species (*Zhang & Archbold, 1993*). Turkey leads Europe in the annual strawberry production by 440,968 tons of yield (*Celiktopuz et al., 2021*). With thick leaves and cuticles, variances in stomatal conductance, and more root development than other Fragaria species, it has been discovered that these species provide better osmotic regulation (*Zhang & Archbold, 1993*). While the fruits of the 279/4 and 279/5 coded genotypes exposed to restricted watering conditions showed no significant change, the fruits of the 253/29 coded genotype showed a substantial decline (*Giné-Bordonaba & Terry, 2016*). Similarly, *Grant et al. (2010)* reported that 10 different strawberry cultivars responded differently based on the severity of water stress. Moreover, *Kapur et al. (2018)* investigated the impact of four irrigation (IR) regimes on yield and physiological parameters. Depending on 0.5, 0.75, 1.0, and 1.25 times the pan's evaporation, they designated the irrigation as IR50, IR75, IR100, and IR125 in the trial where irrigation is scheduled according to pan evaporation (Epan). They found that yield declined dramatically at the IR50 irrigation level, but the Rubygem variety's response was similar at other irrigation levels.

The vapor pressure deficit (VPD) is a significant marker of global water supplies and plant water relations, and due to its global temperature-driven rise, it may become even more essential for vegetation dynamics in the next decades (*Grossiord et al., 2020*). Excess water loss from soils results in drying of terrestrial surfaces and plant water stress when VPD is high (*Dai, Edwards & Ku, 1992*). Furthermore, the method of determining the CWSI for each plant by taking some psychrometric measurements and the difference between the plant crown and the air temperature is one of the most important methods of predicting when and how much plants should be watered. *Sezen et al. (2014)* also stated that the CWSI is the most extensively used index for measuring plant water stress. In addition, CWSI has been proven to be effective in determining irrigation schedules (*Alderfasi & Nielsen, 2001*; *Yazar et al., 1999*; *Irmak, Haman & Bastug, 2000*; *Ehret et al., 2001*; *Hackl et al., 2012*; *Sezen et al., 2014*; *Kim et al., 2015*; *Colak & Yazar, 2017*; *Li et al., 2019*) and yield (*Howell et al., 1984*; *Abdul-Jabbar et al., 1985*; *Kırnak & Gencoğlan, 2001*; *Kayam & Beyazgül, 2001*) after being tested on a variety of plants. CWSI-based measurements for plant water stress monitoring have become the focus of research over the past 30 years and are now being embedded in remote sensing software. However, *Katimbo et al. (2022)* stated that CWSI should be evaluated together with $g_s$, E and Pn in future studies. Furthermore, *Li et al. (2019)* has pointed out the necessity to make calibrations by determining the real CWSI values with field tests in order to obtain automatic CWSI values according to the conditions of each region.

Although crops with greater IWUE are required for long-term agricultural sustainability (*Cattivelli et al., 2008*; *Grant et al., 2012*), no studies have been found to evaluate with the CWSI for strawberry in the Eastern Mediterranean region, where WS will
be felt most in the future. High IWUE and low CWSI, on the other hand, can be associated with yield. The best variety for a certain environment should have a high IWUE and a low CWSI. As a result, evaluating the CWSI and IWUE performances of genotypes with superior genetic traits and commercial varieties can be a viable technique for determining which varieties are best for the region. Therefore, the goal of this study was to determine the most suitable variety for the region by determining the yield, photosynthetic responses, and CWSI-IWUE relationships of four strawberry genotypes with different characteristics grown under high tunnel in Cukurova conditions at two different irrigation levels. It was also aimed to prepare the irrigation program by making use of CWSI.

## MATERIALS AND METHODS

### Site description

The trial was conducted at the Agronomic Research Area, University of Çukurova, Adana, Turkey (latitude: 36°59′N, longitude 35°27′E, 20 m above sea level) during 2019–2020 experimental year (from 18.09.2019 until 20.06.2020). The 30 cm surface layer of the experimental field soil was well-drained clay loam with a bulk density of 1.6 g cm$^{-1}$, pH of 7.6, and 2.12% organic content. At field capacity and permanent wilting stages, the soil water content is 36% and 16%, respectively. The soils at the site were described by USDA as Xerofluvents of the Entisol order with clay texture (*Dingil et al., 2010*).

The plants were grown in Spanish-style high tunnels that were 6.5 m wide, 2.75 m tall, and 40 m long, with UV, IR, AB, EVA, and LD additive plastic that lasted 36 months. The plants were placed in trapezoidal raised beds with a 0.70 m base long, 0.50 m top width, 0.30 m height, and 0.3 m spacing. Each was mulched with a 0.05 mm thick, two-sided polyethylene mulch cover with a grey upper side and a black underneath and surface drip irrigation was linked along the middle of beds.

The inner of the Spanish type of high tunnel is exposed to higher temperature and relative humidity throughout the growing season than the outer, while receiving less solar radiation, according to data collected from meteorological stations (Table 1).

### Experimental design

Plant materials were chosen from four strawberry genotypes (*Fragaria-ananassa* Duch) with different characteristics in the experiment.

a) Rubygem: This is a popular commercial variety with a pleasant flavor and scent, as well as being a short-day and early variety. This variety with bright red color and large fruit is sensitive to powdery mildew disease but tolerant to Fusarium wilt.

b) Festival: It is an early variety. It has been selected in terms of fruit quality, yield and shelf life. It has a conical fruit shape, the flesh color of the fruit is light red, and the outer color of the fruit is dark and bright red (*Türemiş & Ağaoğlu, 2013*).

c) 33: Fortuna and Kaşka types were crossed to form this cultivar. It is a high-yielding genotype that has gotten a lot of attention because of its enormous yields, especially in June, which is the interim period of the strawberry growing seasons. This variety, which has dark red fruit color close to burgundy, is tolerant to fungal diseases (*Sarıdaş, 2018*).
**Table 1 Monthly weather data during the trial.** Each data point reflects the average value of mentioned climate data.

| Months | Average temperature (°C) | | Average humidity (%) | | Average solar radiation (W/m$^2$) | |
|---|---|---|---|---|---|---|
| | Inner | Outer | Inner | Outer | Inner | Outer |
| September | 28.41 | 26.60 | 61.62 | 60.12 | 11.53 | 20.97 |
| October | 25.12 | 23.56 | 60.87 | 59.30 | 9.26 | 14.71 |
| November | 19.09 | 17.66 | 55.13 | 53.67 | 7.69 | 11.14 |
| December | 13.37 | 12.00 | 74.68 | 73.31 | 4.95 | 6.97 |
| January | 10.94 | 9.68 | 64.51 | 63.26 | 6.56 | 8.99 |
| February | 11.49 | 10.17 | 64.98 | 63.53 | 6.02 | 10.56 |
| March | 16.51 | 15.07 | 66.44 | 64.95 | 7.44 | 14.30 |
| April | 18.98 | 17.56 | 63.23 | 67.80 | 10.41 | 20.41 |
| May | 23.37 | 21.90 | 60.02 | 58.66 | 12.34 | 25.19 |
| June | 26.13 | 24.65 | 67.14 | 65.60 | 12.57 | 25.66 |

d) 59: This variety, which preserves its unique fruit shape throughout the season was created by crossing Fortuna and Sevgi cultivars. This genotype differs in that it produces consistently good yields throughout the season, especially in May and June (*Sarıdaş, 2018*).

The trial was implemented as a 4 × 2 factorial scheme of genotypes and irrigation levels, in a split-plot design with three replicates (blocks). There were 80 plants in each block. The main plot was designed with genotypes, and the sub plots were designed with different irrigation levels. According to *Allen et al. (1998)*, losses water above 20% in strawberry are defined as water stress. In this current study the two irrigation treatments, labeled as IR50 (water stress, WS) and IR100 (well-watered, WW) used varying amounts of water and were 0.5, and 1.00, times the Epan which was designated as crop pan coefficient. The Epan value was calculated using a US Weather Service Class A pan with a standard 120.7 cm diameter and 25 cm depth, which was placed over the crop canopy in the high tunnel's center. The Eq. (1) was used to apply irrigation water:

$$IR = A \times Eo \times P \times Kcp \tag{1}$$

where, IR is the irrigation water amount (m$^3$), A is the area of the plot (m$^2$), Eo is the cumulative free surface water evaporation from Class A pan at irrigation interval (mm), P is the wetted area (%), Kcp is the crop-pan coefficients of 0.5, and 1.00 for different irrigation levels throughout the trial. The plants were subjected to the same amount of irrigation water to adapt to the environment until they had three foliate. Different irrigation water applications began on November 8, 2019, and treatments IR100 and IR50 received a total of 727 and 433 mm of water from the beginning to the end of the trial, respectively. There was also no rain or run-off in the high tunnel, thus the plants were only irrigated with irrigation water.

## Measurements

### IWUE and yield

Mature strawberry fruits were harvested twice a week from February to mid of June. The average weight of the fruits harvested from 10 plants was used to calculate the fruit yield in grams per plant (g plant$^{-1}$) whereas the IWUE (g mm$^{-1}$) was determined by dividing the marketable fruit yields by the total amount of irrigation water used (*Yuan, Sun & Nishiyama, 2004*).

### Photosynthetic responses

Measurements of gas exchange were collected throughout the active harvesting months of March, April, and May. In order to monitor the internal water status of the plants, photosynthetic available radiation (Par) ($\mu$mol m$^{-2}$s$^{-1}$), net photosynthetic rate (Pn) ($\mu$mol m$^{-2}$sec$^{-1}$), stomatal conductance ($g_s$) (mmol m$^{-2}$ sec$^{-1}$) and transpiration rate (E) (mmol m$^{-2}$ sec$^{-1}$) measurements were taken on leaves that are completely sun-facing and newly developed in three plants from each plot, at noon (11:00–13:00) with a leaf CI-340 photosynthesis meter (CID/Bio-Science, Camas, WA, USA).

### CWSI

Canopy temperature (Tc) was monitored using an Everest model 110 hand-held infrared thermometer (IRT), which has a field of view of three different angle and catches radiation in the 8–14 mm waveband. The emissivity adjustment was set to 0.98 when the IRT was used. IRT data were taken at a 30°–40° horizontal angle to ensure that only the crop canopy was visible. The first measurements to determine the Tc values were taken on the 187th dap (day after planting), which is the period when the plant cover percentage is around 85%. Dry and wet bulb temperatures were detected with an aspirated psychrometer, which was placed a height of 1.5 m, representing the high tunnel (in the middle). Whereas the average of the dry-bulb temperature values throughout the measurement period was used to calculate the mean Ta, the mean VPD was determined according to the standard psychrometer equation (*Sezen et al., 2014*). The CWSI was determined using an empirical formula developed by *Idso et al. (1981)* (Eq. (2)):

$$\text{CWSI} = (\text{Tc}-\text{Ta})-(\text{Tc}-\text{Ta})_{UL}/(\text{Tc}-\text{Ta})_{UL}-(\text{Tc}-\text{Ta})_{LL} \qquad (2)$$

where the lower limit (LL) denotes the non-water-stressed baseline and the upper limit (UL) denotes the non-transpiring upper baseline; Tc = canopy temperature (°C); Ta = air temperature (°C). Only data from the WW treatments were used to calculate LL for the canopy–air temperature differential (Tc–Ta) against VPD relationship. LL, which is the assumed limit value of plants that are not transpiring at the potential rate, was measured by the Eq. (3) (*Idso et al., 1981*).

$$\text{Tc} - \text{Ta} = a - b.\text{VPD} \qquad (3)$$

In this regression equation, a: represents the inter sectional value of the line (°C), b: the slope of the line (C kPa$^{-1}$). As reported by *Idso et al. (1981)* canopy temperatures of

**Table 2 Different strawberry genotype and irrigation levels effects on yield (g plant$^{-1}$) and IWUE (g mm$^{-1}$).**

| | Genotype | Irrigation levels | Irrigation × Genotype | Average of genotype |
|---|---|---|---|---|
| Yield | Rubygem | WW | 864.9 | 755.0 AB |
| | | WS | 645.1 | |
| | Festival | WW | 847.8 | 696.4 B |
| | | WS | 545.0 | |
| | 33 | WW | 1,029.4 | 798.3 AB |
| | | WS | 567.2 | |
| | 59 | WW | 1,067.4 | 866.5 A |
| | | WS | 665.6 | |
| | Average of irrigation levels | WW | *952.4 A* | Lsd$_{genotype}$* = 117 |
| | | WS | *605.7 B* | Lsd$_{irrigation}$* = 83 |
| | | | | Lsd$_{irrigation × genotype}$ = N.S |
| IWUE | Rubygem | WW | 1.19 | 1.34 |
| | | WS | 1.49 | |
| | Festival | WW | 1.17 | 1.21 |
| | | WS | 1.26 | |
| | 33 | WW | 1.41 | 1.36 |
| | | WS | 1.31 | |
| | 59 | WW | 1.47 | 1.50 |
| | | WS | 1.54 | |
| | Average of irrigation levels | WW | *1.31* | Lsd$_{genotype}$ = N.S |
| | | WS | *1.40* | Lsd$_{irrigation}$ = N.S |
| | | | | Lsd$_{irrigation × genotype}$ = N.S |

**Notes:**
* $P \leq 0.05$.
N. S., Not Significant, Rubygem, Festival, 33, and 59 represent the name of the genotypes. WW refers for well-watered, while WS refers for water stress. separate letters represent the differences between the averages.

completely stressed plants were measured to detect UL. In addition, hourly measurements were taken between 08:00–17:00 and compared to determine the CWSI × g$_s$ relationship at the end of the growing season.

## Statistical analysis

The trial was implemented as a 4 × 2 factorial scheme of genotypes and irrigation levels (Table 2) in a split-plot design with three replicates (blocks). The trial was, also, implemented as a 4 × 2 × 3 factorial scheme of genotypes, irrigation levels, and period (Table 3) in a split split-plot design with three replicates. There were 80 plants in each block. The main plot was designed with genotypes, and the sub plots were designed with different irrigation levels. In the JMP 8.1 statistical analysis package program, the variance analysis was performed. The differences between the averages were compared with the LSD test at the 5% significance level ($P \leq 0.05$). Regression analysis was also used to determine the relationships between some important parameters.

**Table 3 Different strawberry genotypes and irrigation levels effects on photosynthetic responses during the active harvesting period.**

| | Genotype | Irrigation | Period | | | Irrigation × Genotype | Average of genotype |
|---|---|---|---|---|---|---|---|
| | | | March | April | May | | |
| Par ($\mu$mol m$^{-2}$s$^{-1}$) | Rubygem | WW | 192 | 744 | 733 | *556 c* | 446 C |
| | | WS | 95 | 462 | 448 | *335 f* | |
| | Festival | WW | 165 | 727 | 720 | *537 d* | 436 D |
| | | WS | 109 | 452 | 446 | *335 f* | |
| | 33 | WW | 150 | 780 | 774 | *568 b* | 456 B |
| | | WS | 81 | 487 | 466 | *345 f* | |
| | 59 | WW | 206 | 797 | 789 | *597 a* | 479 A |
| | | WS | 95 | 498 | 492 | *362 e* | |
| | Average of period | | 136 C | 618 A | 608 B | | |
| | Average of irrigation | WW | | *565 A* | | Lsd$_{irrigation}^* = 5.9$ | Lsd$_{genotype}^* = 8.3$ |
| | | | | | | Lsd$_{period}^* = 7.2$ | Lsd$_{genotype \times period}^* = 14$ |
| | | WS | | *344 B* | | Lsd$_{irrigation \times genotype}^* = 12$ | Lsd$_{irrigation \times period}^* = 10$ |
| | | | | | | Lsd$_{irrigation \times genotype \times period} = $ N.S | |
| Pn ($\mu$mol m$^{-2}$sec$^{-1}$) | Rubygem | WW | 7.1 | 12.8 | 12.2 | *10.7 b* | 8.8 B |
| | | WS | 6.0 | 8.2 | 6.3 | *6.8 e* | |
| | Festival | WW | 7.3 | 12.9 | 12.4 | *10.9 b* | 8.9 B |
| | | WS | 6.3 | 8.2 | 6.2 | *6.9 de* | |
| | 33 | WW | 8.4 | 13.9 | 13.4 | *11.9 a* | 9.6 A |
| | | WS | 6.0 | 8.9 | 6.7 | *7.2 cd* | |
| | 59 | WW | 7.8 | 13.8 | 13.3 | *11.6 a* | 9.5 A |
| | | WS | 5.7 | 8.8 | 7.8 | *7.5 c* | |
| | Average of period | | 6.8 C | 11.0 A | 9.8 B | | |
| | Average of irrigation | WW | | *11.3 A* | | Lsd$_{irrigation}^* = 0.18$ | Lsd$_{genotype}^* = 0.26$ |
| | | | | | | Lsd$_{period}^* = 0.22$ | Lsd$_{genotype \times period}^* = 0.45$ |
| | | WS | | *7.1 B* | | Lsd$_{irrigation \times genotype}^* = 0.37$ | Lsd$_{irrigation \times period}^* = 0.32$ |
| | | | | | | Lsd$_{irrigation \times genotype \times period} = $ N.S | |
| g$_s$ (mmol m$^{-2}$ sec$^{-1}$) | Rubygem | WW | 232 m | 608 d | 581 e | *474 c* | 383 D |
| | | WS | 181 o | 396 g | 302 j | *293 f* | |
| | Festival | WW | 272 k | 612 d | 583 e | *489 b* | 407 C |
| | | WS | 235 m | 401 g | 343 hı | *326 e* | |
| | 33 | WW | 252 l | 684 a | 664 b | *533 a* | 430 A |
| | | WS | 231 m | 447 f | 305 j | *328 d* | |
| | 59 | WW | 269 k | 673 ab | 644 c | *529 a* | 413 B |
| | | WS | 207 n | 335 ı | 352 h | *298 f* | |
| | Average of period | | 235 C | 519 A | 471 B | | |
| | Average of irrigation | WW | | *506 A* | | Lsd$_{irrigation}^* = 3.5$ | Lsd$_{genotype}^* = 4.9$ |
| | | | | | | Lsd$_{period}^* = 4.3$ | Lsd$_{genotype \times period}^* = 8.6$ |
| | | WS | | *311 B* | | Lsd$_{irrigation \times genotype}^* = 7.0$ | Lsd$_{irrigation \times period}^* = 6.1$ |
| | | | | | | Lsd$_{irrigation \times genotype \times period}^* = 12$ | |

(Continued)

| Table 3 (continued) | | | | | | | | |
|---|---|---|---|---|---|---|---|---|
| | Genotype | Irrigation | Period | | | Irrigation × Genotype | Average of genotype | |
| | | | March | April | May | | | |
| E (mmol m$^{-2}$ sec$^{-1}$) | Rubygem | WW | 1.40 | 3.60 | 3.40 | *2.80 c* | 2.19 D | |
| | | WS | 1.10 | 2.10 | 1.53 | *1.58 g* | | |
| | Festival | WW | 1.47 | 3.65 | 3.47 | *2.86 b* | 2.25 C | |
| | | WS | 1.17 | 2.13 | 1.63 | *1.64 f* | | |
| | 33 | WW | 1.55 | 3.90 | 3.77 | *3.07 a* | 2.40 B | |
| | | WS | 1.03 | 2.40 | 1.73 | *1.72 e* | | |
| | 59 | WW | 1.55 | 3.95 | 3.77 | *3.09 a* | 2.44 A | |
| | | WS | 1.10 | 2.43 | 1.83 | *1.79 d* | | |
| | Average of period | | 1.30 C | 3.02 A | 2.64 B | | | |
| | Average of irrigation | WW | | *2.95 A* | | Lsd$_{irrigation}$* = 0.02 | Lsd$_{genotype}$* = 0.03 | |
| | | | | | | Lsd$_{period}$* = 0.03 | Lsd$_{genotype \times period}$* = 0.06 | |
| | | WS | | *1.68 B* | | Lsd$_{irrigation \times genotype}$* = 0.05 | Lsd$_{irrigation \times period}$* = 0.04 | |
| | | | | | | Lsd$_{irrigation \times genotype \times period}$ = N.S | | |

**Notes:**
* $P \leq 0.05$.
N. S., Not Significant, Rubygem, Festival, 33, and 59 represent the name of the genotypes. WW refers for well-watered, while WS refers for water stress. Separate letters represent the differences between the averages.

# RESULTS

## Yield and IWUE

The highest yield value in the experiment resulted from a WW-59 interaction with 1,067 g plant$^{-1}$, while the lowest resulted from a WS-Festival interaction with 545 g plant$^{-1}$. The differences between both genotype and irrigation applications were found to be statistically significant ($P \leq 0.05$). The genotype 59 had the highest average yield, while the Festival had the lowest. The WW (IR100) application produced an average of 952.4 g plant$^{-1}$ strawberry yield, while the WS (IR50) produced an average of 605.7 g plant$^{-1}$, indicating that WS significantly reduced the yield.

The highest average IWUE obtained from the WS-59 interaction (1.54 g mm$^{-1}$), while the lowest obtained from the WW-Festival interaction. Despite the fact that genotype 59 had the highest average IWUE (1.50 g mm$^{-1}$), the differences between genotypes were not statistically significant. In addition, the WW application had an average of 1.31 g mm$^{-1}$ IWUE and the WS application had 1.40 g mm$^{-1}$, but these differences were not significant as well. Although the differences in irrigation × genotype interactions for both parameters in the study were not statistically significant, genotype 59 had the highest yield and IWUE when WS conditions were considered, demonstrating that the genotypes had varying drought tolerance (Table 2).

## Photosynthetic responses

All photosynthetic parameters evaluated in the experiment had statistically significant variations in irrigation, genotype, period, genotype × period, irrigation × genotype, and irrigation × period ($P \leq 0.05$). The interaction of irrigation × genotype × period was found

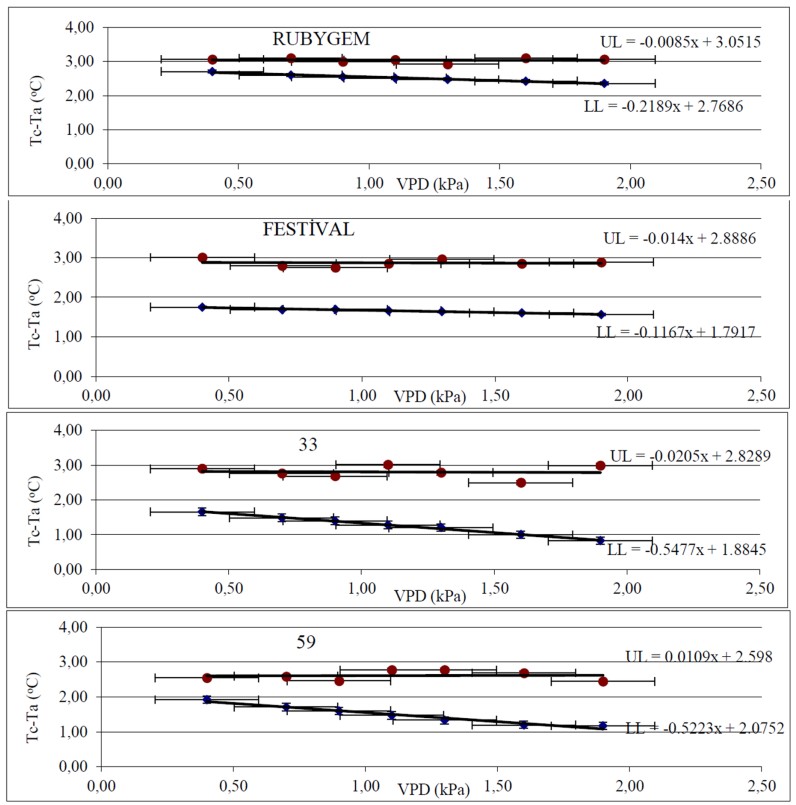

**Figure 1 Canopy temperature–air temperature (Tc–Ta) × vapor pressure deficit (VPD) regression graphs of different strawberry genotypes.**

to be statistically insignificant in the other examined parameters (exception of $g_s$). Whereas genotype 59 had the greatest E, genotype 33 had the highest $g_s$. Apart from these, genotypes 33 and 59 were found to be in the highest statistical group in terms of Pn. The WS application reduced all values by a significant amount in all of the parameters studied, indicating that plants subjected to water deficiency have a lower photosynthetic performance, which may be partly ascribed to lower Par incident in leaf blade.

Considering the irrigation × genotype interaction, it was discovered that 59 and 33 applications were found together under WW conditions in the most statistically significant group (except for Par). The interaction WW-59 produced the highest Par. Furthermore, when all parameters were assessed as a period, the greatest values were observed in April (Table 3). Considering merely drought conditions (WS), genotype 59 had the greatest performance in all other parameters except $g_s$ (genotype 33), demonstrating that genotypes have varied photosynthetic responses under water stress.

## CWSI

According to the VPD × Tc–Ta regression analysis, the LL without water stress and the UL where the plant is completely water stressed equations were determined for each genotype and displayed in Fig. 1. In the absence of water stress, the equations $LL_{Rubygem} = -0.2189x + 2.7686$, $LL_{Festival} = -0.1167x + 1.7917$, $LL_{33} = -0.5477x + 1.8845$, and $LL_{59} = -0.5223x +$

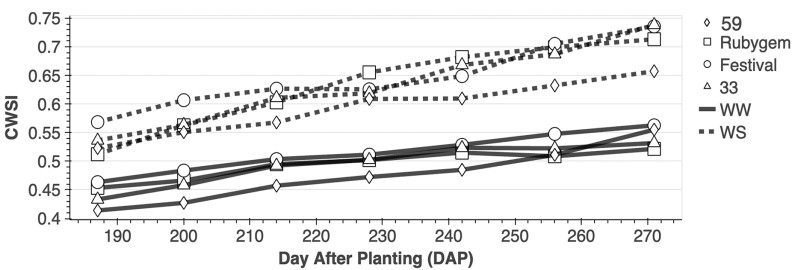

**Figure 2 CWSI changes of strawberry genotypes at different irrigation levels.**

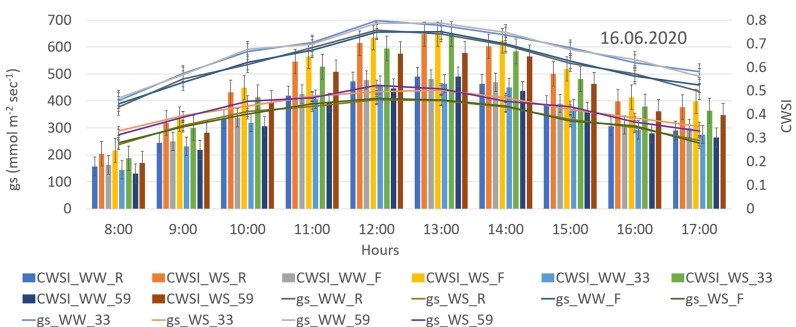

**Figure 3 Hourly CWSI plotted together with $g_s$ values of genotypes under different irrigation levels.**

2.0752 were obtained. The fact that different genotypes with the same circumstances have different LL equations proves that even within the same plant species, genetic differences affect LL. Furthermore, all LL equations had positive inter sections, showing that water vapor transport from the leaf to the atmosphere continues even when the atmosphere is totally saturated (VPD = 0). Considering the UL equations of the genotypes, it was discovered that the Tc–Ta differences varied between the genotypes (the slopes in the equations were neglected because they were too low). The highest Tc–Ta was found in $UL_{Rubygem}$ (3.05), while the lowest was found in $UL_{59}$ (2.60). According to the UL equations representing extreme water stress conditions, the most tolerant genotype was 59, while Rubygem was the least tolerant.

The lowest CWSI (0.41) was achieved in dap 187 from the WW-59 interaction, while the highest (0.74) was found in dap 271 from the WS-33 interaction (Fig. 2). The average CWSI values, at the end of the study, were 0.47 (WW-59), 0.49 (WW-33), 0.49 (WW-Rubygem), 0.51 (WW-Festival), 0.59 (WS-59), 0.63 (WS-33), 0.63 (WS-Rubygem) and 0.65 (WS-Festival) (Fig. 3).

It has been observed that CWSI and soil moisture level have a very strong negative relationship. The CWSI values of genotypes with different characteristics under the same conditions differed as well, which can be explained by the inverse relationship between soil moisture and CWSI and the genotypes' different drought tolerance. Therefore, the WW-59 interaction provided the lowest average CWSI value, whereas the WS-Festival interaction provided the highest average CWSI. Furthermore, the average CWSI values of WW

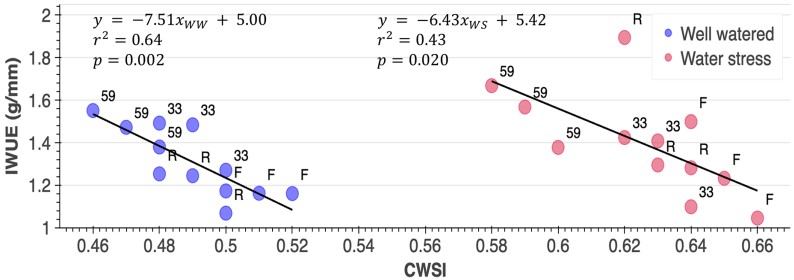

**Figure 4** Responses of genotypes (Rubygem, R; Festival, F; 59, 33) to the CWSI (crop water stress index) × IWUE (irrigation water use efficiency) (g mm$^{-1}$) relationship under different irrigation levels.

application were determined to be 0.49, and the average CWSI value of WS application was determined to be 0.63, indicating that strawberry irrigation management can be managed under the high tunnel in Mediterranian using CWSI values between 0.49 and 0.63.

At the harvest period, the combined diurnal CWSI and $g_s$ graph revealed that both CWSI and $g_s$ values rose until 13.00 h and then tended to decline (Fig. 3). The positive relation between stomatal conductivity and CWSI is remarkable. Moreover, the explanation for this is that stomatal openings reach the maximum in the afternoon and subsequently decrease in the evening to protect plants agaits excessive water loss, altering the VPD and causing fluctuations in the CWSI. Furthermore, when CWSI × $g_s$ was evaluated as genotype, it was discovered to have a negative correlation, indicating that plants with a high CWSI restrict their stomata to reduce evaporation. This clearly indicates that the genotypes Festival and Rubygem (had lowest $g_s$ and highest CWSI) are the most sensitive genotypes to the research conditions.

The relationship between IWUE and CWSI$_{average}$ is found to be negative in this study (Fig. 4). The genotype with the highest IWUE and lowest CWSI is the most drought resistant in the WS conditions. In this context, the genotype with the highest IWUE (1.54) and lowest CWSI (0.59) was genotype 59 under WS conditions. Besides, the genotype Festival was shown to be the most drought sensitive.

## DISCUSSIONS

### Yield and IWUE

WS considerably reduced (36.4%) strawberry yield (Table 2), as previously reported in other researchers (*Yuan, Sun & Nishiyama, 2004*; *Liu et al., 2007*; *Klamkowski & Treder, 2008*; *Grant et al., 2010*; *Ghaderi, Normohammadi & Javadi, 2015*; *Adak, Gübbük & Tetik, 2018*; *Sarıdaş et al., 2021*). Furthermore, the cultivars reactions to the WS application differed, with yield values ranging from 866.5 g plant$^{-1}$ (genotype 59) to 696.4 g plant$^{-1}$ (genotype Festival). Similarly, *Grant et al. (2010)* found that 10 different strawberry cultivars responded differently to 30% limited irrigation, with output decreases likely related to the severity of WS. *Adak, Gübbük & Tetik (2018)* reported that deficit irrigated (half of the control group) strawberries had a lower yield of 63.6% in total yield. The authors also determined that Albion and Rubygem genotypes were more tolerant to

WS than genotype Amiga. Even though the Rubygem, which is supposed to be more drought resistant, was employed in this study, genotypes 59 and 33 were discovered to be more drought resistant. According to *Klamkowski & Treder (2008)*, three different strawberry cultivars (Elsanta, Elkat, and Salut) reacted differently to WS (half of the control group), with Elkat yielding the lowest while Elsanta yielding the most.

IWUE has been used in some studies on strawberry genotype water interactions and drought tolerance in a variety of climates (*Yuan, Sun & Nishiyama, 2004*; *Klamkowski & Treder, 2008*; *Grant et al., 2010*; *Ghaderi & Siosemardeh, 2011*; *Klamkowski, Treder & Wojcik, 2015*; *Ferri et al., 2016*). The IWUE for the WW application, in the current study, was 1.31 g mm$^{-1}$, while the IWUE for the WS application was 1.40 g mm$^{-1}$. The IWUE increased as the amount of irrigation water used decreased. *Yuan, Sun & Nishiyama (2004)* emphasized a similar result. They obtained, also, the best yield at IWUE 1.63 g mm$^{-1}$ conditions after a trial in which they tried three different irrigation levels in strawberries. The optimal IWUE value was 1.47 g mm$^{-1}$ in the current investigation since the highest yield value obtained from WW-59 interaction (1,067 g plant$^{-1}$). Moreover, the genotypes in our investigation had average IWUE values ranging from 1.21 to 1.50 g mm$^{-1}$. Similarly, *Grant et al. (2010)* found that the IWUE values of 10 different strawberry cultivars varied, and that the Hapil and Totem cultivars were more resistant to water stress than the others. *Ferri et al. (2016)* found that IWUE in strawberries varied greatly depending on cultivar, and *Klamkowski & Treder (2008)* indicated that among three strawberry cultivars, the Elsanta cultivar had the greatest IWUE values under water deficit conditions.

## Photosynthetic responses

All photosynthetic parameters are strongly influenced by the especially irrigation, period, genotype and irrigation × genotype (Table 3). *Lawlor (2002)* and *Yordanov, Velikova & Tsonev (2000)* both confirmed that WS had a significant impact on photosynthetic capacity. Besides, Manzanar Alto from South American *F. chiloensis* lines had similar E with commercially grown strawberries, but *F. chiloensis* types from North America use considerably less water than *F. × ananassa* (*Grant et al., 2012*). In the same study, significant reductions in $g_s$ and Pn were detected under limited irrigation conditions, although at different levels among genotypes. Similarly, according to *Mao et al. (2009)*, a lack of water in the soil reduced Pn, $g_s$, and E. Consistent with previous studies, in the current study, WS caused decreases in Pn, $g_s$, and E by 37%, 39%, and 43%, respectively, indicating that plants subjected to water deficiency have a lower photosynthetic performance. Moreover, under mild and moderate stress, some experts believe that stomatal closure is the principal predictor of decreased photosynthesis and yield (*Klamkowski & Treder, 2008*). The genotype with the second highest average $g_s$ value produced the highest yield in the current study (genotype 59). This difference is assumed to be related to the genotype's smaller leaves compared to other genotypes (Fig. 5), and the genotype's attempt to adapt to the environment by reducing $g_s$ compared to genotype 33 under water stress. *Mao et al. (2009)*, also, found that as the level of WS increased, photosynthetic activities were reduced more severely and to varying degrees in different genotypes. *Klamkowski, Treder & Wojcik (2015)*, also, pointed out that the rate of gas

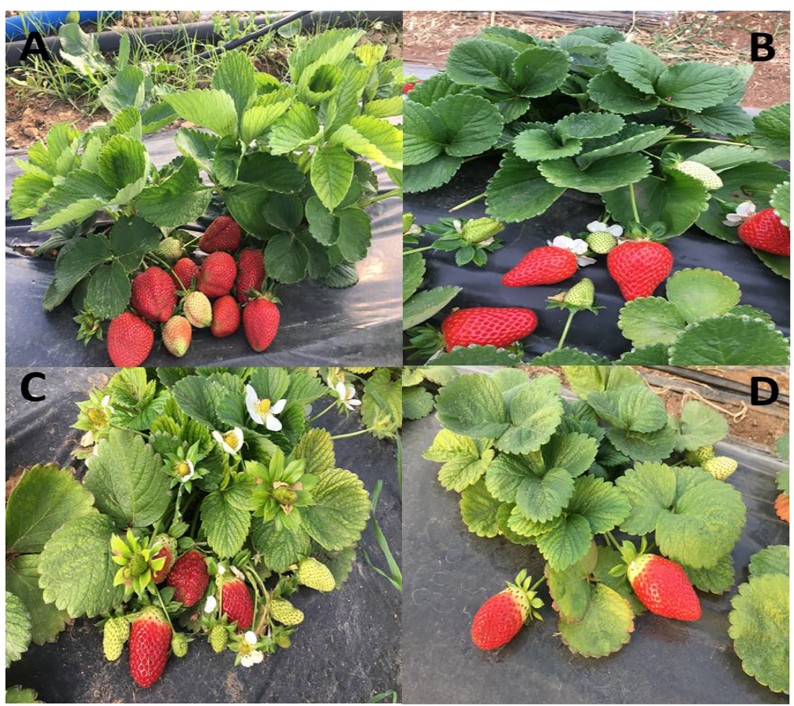

**Figure 5** Images of different genotypes (Rubygem (A); Festival (B); 33 (C); 59 (D)).

exchange lowered with various levels in all cultivars as WS increased. Similarly, genotypes showed varying drought tolerance responses in the current study. The genotypes 59 had the strongest photosynthetic performance overall (except $g_s$), especially when compared to drought conditions. Furthermore, there was an increase in all the parameters examined in the study from the first period of measurement to the mid-period, and there were significant decreases in the last harvest period (Table 3). These findings support *Carlen, Potel & Ancay (2009)*'s discovery that photosynthetic parameters increased from the beginning to the middle of harvest, then fell until the end of the experiment due to water demand during the harvest period.

## CWSI

Different genotypes have different LL and UL equations (Fig. 1). When compared to a reference temperature (air temperature), vegetation temperature measured radiometrically is a useful predictor of water stress (*Jackson et al., 1983*). In sorghum, *Stricevic & Caki (1997)* discovered a strong link among soil water content, leaf water potential, and Tc–Ta interactions. *Smith, Prathapar & Barrs (1989)* found that combining statistical analysis and plant surface temperature, soil water level may be calculated. In the current study, genotype Rubygem had the highest Tc–Ta, whereas genotype 59 had the lowest, indicating that yield, Pn, and E have a strong negative relation with Tc–Ta. Furthermore, the most drought tolerant genotype was 59 under WS condition as well, indicating that genotype 59 has better ability to regulate leaf temperature under WS conditions. Drought tolerance

differences are thought to due to genetic characteristics. Similarly, *Reginato (1983)* stated that plants with small leaves are more affected by temperature.

Strong negative relationship detected between irrigation levels and CWSI in this study. Similar results were obtained by *Sadler et al. (2000)* in corn, and by *Orta, Erdem & Erdem (2001)* in sunflowers. WW-59 interaction provided the lowest average CWSI value, whereas WS-Festival interaction provided the highest average CWSI when compared interactions (Fig. 4). Moreover, the genotype with the highest IWUE (1.54) and lowest CWSI (0.59) was genotype 59 under WS conditions, indicating that genotype 59 is the most drought resistant in the current study and proves that genetic differences affect drought tolerance (Figs. 2 and 4). Similarly, *Grant et al. (2012)* found that CWSI values of genotypes varied significantly depending on irrigation amounts. In the same research, scientists also discovered a statistically significant negative correlation between $g_s$ and CWSI for each genotype. The 59 and 33 genotypes with the highest $g_s$ values were consistently determined to have the lowest CWSI in the current study. Moreover, the combined diurnal CWSI and $g_s$ graph revealed that both CWSI and $g_s$ values rose until 1:00 pm, after which they started to decline (Fig. 3). However, in accordance with other data, it was determined that the genotypes with the highest $g_s$ had the lowest CWSI. Similarly, *Ehrler et al. (1978)* found out that Tc–Ta increased rapidly after morning hours in dry soil conditions, then gradually decreased after 14:00. Furthermore, several researches (*Jackson et al., 1983*; *Ehrler et al., 1978*; *Koksal et al., 2006*) suggest that the best period to measure plant surface temperature and monitor water stress is between 1:00 and 2:00 p.m. In this context, the hourly data collected in our study are consistent with the literature, and it is reasonable to conclude that the best time to measure leaf surface temperature of strawberries is around 1:00 pm.

## CONCLUSION

The future of strawberry production depends on productive, high quality and drought tolerant varieties. So, the goal of this study was to determine the most suitable variety for the region by determining the yield, photosynthetic responses, and CWSI-IWUE relationships of strawberry genotypes with four different characteristics grown under high tunnel in Cukurova conditions at two different irrigation levels. It was also aimed to prepare the irrigation program by making use of CWSI.

As a result of the research, the equations $LL_{Rubygem} = -0.2189x + 2.7686$, $LL_{Festival} = -0.1167x + 1.7917$, $LL_{33} = -0.5477x + 1.8845$, and $LL_{59} = -0.5223x + 2.0752$ were obtained. The fact that different genotypes with the same circumstances have different LL equations proves that even within the same plant species, genetic differences affect LL. In this research, genotype Rubygem had the highest Tc–Ta, whereas genotype 59 had the lowest, indicating that genotype 59 has better ability to thermoregulate leaf temperatures. Moreover, yield, Pn, and E were found to have a substantial negative relationship with Tc–Ta.

WS reduced yield, Pn, $g_s$, and E by 36%, 37%, 39%, and 43%, respectively, whereas it increased CWSI (22%) and IWUE (6%). Besides, the optimal time to measure leaf surface temperature of strawberries is around 1:00 pm and strawberry irrigation management

might be maintained under the high tunnel in Mediterranean utilizing CWSI values between 0.49 and 0.63.

Although genotypes had varying drought tolerance, the genotype 59 had the strongest yield and photosynthetic performances overall (except $g_s$), especially when compared to drought conditions. Furthermore, 59 had highest IWUE and lowest CWSI in the WS conditions, proving to be the most drought tolerant genotype in this research.

## ACKNOWLEDGEMENTS

I sincerely appreciate the help provided by Assoc. Prof. Dr. Mehmet Ali Saridas with crop husbandry. I also acknowledge the technical support provided by Prof. Dr. Guy Fipps and Assoc. Prof. Dr. Burcak Kapur.

### Funding

The authors received no funding for this work.

### Competing Interests

The authors declare that they have no competing interests.

### Author Contributions

- Eser Celiktopuz conceived and designed the experiments, performed the experiments, analyzed the data, prepared figures and/or tables, authored or reviewed drafts of the article, and approved the final draft.

### Data Availability

The raw data is available in the Supplemental File.

### Supplemental Information

Supplemental information for this article can be found online at http://dx.doi.org/10.7717/peerj.14972#supplemental-information.

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
