# Peer review of "Determination of drought tolerance of different strawberry genotypes"

_PeerJ, doi:10.7717/peerj.14972_

## Round 0.1 · original submission · Major Revisions

In addition to the Reviewers’ comment below, I have the following comments:

The most crucial missing information of your work is the replication of plants. Please specify how many plants per block you used in your experiment.

The second important issue is the statistical approach of your data. Please give more information about the specific tests you used in the analysis of variance.

The Abstract should be re-written, with a different form: first a introductory sentence to the subject, then the objectives of the study, third the treatments and parameters measured, then the main results and finally the conclusions.

Figures should be improved:
A) please include units and error bars
B) Re-write the captions since a Figure caption is self-explanatory text, and particularly consider the following:
1) they should be more explanatory and
2) explain the abbreviations (e.g. DAP)

Figure 4: I cannot see any CWSC x IWUE relationship, but only two separate figures presented together. A relationship of such a kind needs statistical testing of correlation or regression or other.

·

Basic reporting

Clear and unambiguous, professional English used throughout
In general, the author has prepared the manuscript in a sufficiently clear and unambiguous, technically correct English. Nevertheless, some minor corrections are needed so I recommend a thorough revision of the paper regarding this issue. This reviewer has made some corrections that are indicated here just as examples of the language revision to be made (these do no pretend to be exhaustive):
Line 46: ‘global warming due to climate change’
Line 50: it is preferable ‘In this context,’ instead of ‘In this regard, ... ‘
Line 106: first ‘width’ in the text line should be substituted by ‘long’
Line 124: add ‘of’ after ‘period’
Line 205: correction, ‘harvesting’
Line 243: correct ‘relation’ by ‘relationship’ or ‘correlation’
Line 304: substitute ‘is’ by ‘are’, and ‘WS’ by ‘irrigation’
Line 319: correct ‘producing less’ by ‘reducing’, it has no sense ‘producing less gs’
Line 351: delete ‘that’
Three last important remarks regarding expressions and terminology:
1) The author used several times the expression “of four strawberry genotypes with four different characteristics” (lines 23-24; line 91; lines 365-366), which does not make sense. A more proper expression would be “of four strawberry genotypes with divergent (or different) characteristics”
2) The author is confusing species with variety in some instances: lines 227-228 and 372. It is important to remark that the author is studying four genotypes of the same strawberry species (Fragaria ananassa Duch) (line 116), so what it should be written in those lines is: “even within the same plant species genetic differences affect LL”.
3) Use and definition of abbreviations: author makes use of a remarkable number of abbreviations, undoubtedly needed in this type of article, but sometimes they are not defined the first time they appear in the text. This occurs fundamentally in Abstract but also in other parts of the manuscript (e.g. line 63 with VPD abbreviation). This reviewer urges to correct this issue and highly recommends the author to insert a list of abbreviations used in the paper, given the so high number of them (IR, PAR, Pn, gs, E, LL, Tc, Ta, DAP, etc.). One last remark regarding this issue: CSWI is defined in Abstract as ‘crop water stress index’, so in Introduction (line 67) this definition should be maintained (in such line it is defined as plant water stress index), which is in accordance with the acronym.
Literature references, sufficient field background/context provided.
The author has made use of the relevant literature and provide sufficient field background to contextualize the research work presented, fitting it the general field of research of crop plants tolerance to drought.
There is only an issue regarding this aspect that author may improve and it is in Introduction section. He should add some information about the agricultural and socio-economic importance of strawberry, supported by some agronomic data (production, cultivated area, yield) perhaps indicating the position of Turkey in this regard in the world (as these research work has been performed in Turkey). Also it would be good if something is said about the contribution of strawberry to the human diet, that is to say, the beneficial compounds it contains for human health.
Professional article structure, figures, tables. Raw data shared.
The article structure (abstract, keywords, introduction, material & methods, results, discussions, conclusions, acknowledgements and references) is the standard one for a scientific article and in general it follows the guidelines of the journal. Only regarding Abstract the author has not strictly followed the structure required by the journal (the abstract should be subdivided in Background, Methods and Results according to Author Instructions of the journal).
All figures and tables shown in the paper are relevant to the content of the article, but some of them need some improvements in format and/or labelling. This reviewer is going to present his comments at this regard:
Figure 1: y-axis in the graphic at the bottom end needs to be put in the same scale as the rest (with 0.00/1.00/2.00/4.00 subdivisions)
Figure 2: Increase the length of the y-axis for the reader to be able to discriminate the evolution of the CWSI parameter along the period of cultivation for each genotype and irrigation treatment, making use of a more precise scale (0.10-units subdivision). In this case this reviewer thinks it is best to use another labelling of the data points in the graphic for sake of clarity and appeal (for example making use of black and white squares for Rubygem, black and white circles for Festival, black and white triangles for 33 and black and white diamonds for 59, for WW and WS treatments, respectively, with continuous black lines for WW treatment and broken black lines for WS for evolutions)
Figure 3: It is highly recommendable to increase the length of the y-axis for the same reason as in previous figure, considering more divisions in the scale of gs (e.g. 100-units)
Table 1: Decimal point is missing in outer average humidity in September column
Table 2: Regarding Yield parameter: at the column Irrigation x Genotype no letters are shown between treatments (WW vs WS) for each genotype, so it is supposed that no statistically significant differences have been detected. Looking at the values it seems a bit surprising given the so high apparent differences between treatments in each genotype but still it seems LSD for irrigation x genotype interaction does not show significant differences. This needs to be verified. Regarding IWUE parameter no statistically significant differences are found in any of the two factors (genotype & irrigation), and in the interaction of both, so here it is clear the absence of letters.
Table 3: This is the table that most improvements are needed: (1) Units used for each photosynthetic parameter should be presented; (2) uniformity in letters format for statistically significant differences is required: In Irrigation x Genotype column letters appear in lowercase for PAR parameter, while for the rest of photosynthetic parameters these are uppercase; (3) no letters appear for PAR, Pn and E parameters in columns for period of measurement (March-April-May), only for gs there are letters denoting statistically significant differences, but LSD results for period factor indicate there are indeed statistically significant differences for this factor in every parameter measured (7.2 for PAR, 0.22 for Pn, 4.3 for gs and 0.03 for E).
For all figures and tables: in figure legends and table captions abbreviations used should be defined.
This reviewer thinks all appropriate raw data have been made available in accordance with the Data Sharing policy of the journal.
Self-contained with relevant results to hypotheses.
In the view of this reviewer the submitted manuscript is ‘self-contained,’ representing an appropriate ‘unit of publication’, and it includes all results relevant to approach the goal of the research work.

Experimental design

Original primary research within Aims and Scope of the journal. Research question well defined, relevant & meaningful. It is stated how research fills an identified knowledge gap.
The manuscript clearly defines the goal of the research work that describes, at the end of Introduction section: it is the identification among a group of four strawberry genotypes, characterized by different assets (quality traits, yield, shelf life ability, described in lines 118-127) the most tolerant one to water deficit, imposed by reducing to a half the standard irrigation (100% pan evaporation, Epan). Perhaps in Abstract it could be better emphasized, as it is somewhat confusingly stated in lines 21-25, although it is better defined at the end of it, in lines 35-39, but as final results obtained, not as a proper statement of the research objective.
This is an applied research goal, that is, it does not try to answer to a fundamental research question, in order to fill a gap of fundamental knowledge. But this aspect does not decrease at all the interest of the research in view of this reviewer, as the identification of drought tolerant crop varieties, in this case strawberry, is a critical objective in applied research in agricultural sciences, particularly in the current climate change scenario.
Rigorous investigation performed to a high technical & ethical standard.
This reviewer finds that the research has been conducted rigorously and to a high technical standard, and in conformity with ethical standards for conducting research. No other ethical issues apply in this research work (no use of transgenic or indigenous protected plant material, no use of hazardous growth conditions in the field, …).
Methods described with sufficient detail & information to replicate.
Methods have been described with sufficient information to be reproducible by another investigator.

Validity of the findings

Impact and novelty not assessed. Meaningful replication encouraged where rationale & benefit to literature is clearly stated.
This is a meaningful research work, clearly described and, given its goal, already assessed in previous criteria (Experimental Design), it is not a mere uninteresting replication.
All underlying data have been provided; they are robust, statistically sound, & controlled.
The data presented in the manuscript are robust, statistically sound and controlled and clearly support the conclusions and these are provided as supplemental files.
The only point regarding this issue that needs improvement is formatting and labelling of certain figures and tables, that this reviewer has already detailed in evaluation of first criteria (Basic Reporting).
Conclusions are well stated, linked to original research question & limited to supporting results.
In view of this reviewer the conclusions of the research work are clearly stated and undoubtedly linked to the goal of it and clearly delimited and supported by the results presented. The correlations found, particularly that of CWSI and gs described in lines 257-260, 351-356 and in figure 3, are robustly supported by the results of a well-controlled experiment.

Additional comments

Other comments this reviewer would like to add and does not know very well where to fit them in previous sections are the following:
It is best to place Table 1 at the end of subsection of Material and Methods ‘Site description’, just after the last paragraph, where environmental conditions are mentioned. At the end of such paragraph this table 1 should be cited.
First paragraph of Discussions (lines 274-286): it is important here to, at least briefly, refer to conditions of water stress treatments applied by the authors cited, as most probably they differ from the one used in these research work. Differing water deficit conditions may very well explain differences found in plant responses, even for the same species and cultivar, so trying to compare results may be really hard.
Lines 316-319 and 338-342: Author considers here that in case of the strawberry genotype showing the highest tolerance to water stress, cultivar ‘59’, this tolerance is related to a particular phenotypic trait, the leaf size, in which differs from the other three genotypes. In order for the reader to be able to appreciate this apparently so important morphological divergence of cultivar 59 with respect to the other three, the introduction in the manuscript of a figure with pictures of the plant and leaf of each cultivar would be most welcomed.

Reviewer 2 ·

Basic reporting

In this manuscript titled “Determination of drought tolerance of different strawberry genotypes”, the author reported the results from a field trial of different strawberry genotypes. The author made substantial efforts in conducting the field trial and quantifying the parameters. However, the manuscript is lacking some important details. To improve the quality of the manuscript, the following concerns need to be addressed.
1. Please provide phenotypic pictures of the four strawberry genotypes, under the different experimental conditions, as this is important result to support the conclusion of this study.
2. In the abstract, it is not appropriate to use abbreviations before defining them first. Please use the full name in the abstract, for example, “LL” and “x” in the equation, “Tc-Ta”, “Par, Pn, gs and E”.
3. The author needs to explain the definition of IR in the introduction, not just in the experimental design. And the author needs to explain why is IR50 considered water stress (WS) condition for strawberries. The author needs to refer to previous studies in more details to introduce the current knowledge about strawberry growth under different watering conditions.
4. For the four strawberry genotypes tested in this study, please expand the description of the common features and differences in Line 118-127.
5. It would be helpful if the author could include a satellite map marked with grids to illustrate the field design described in Line 128-135.
6. It seems like only 10 plants are included in each condition. 10 plants do not seem to be sufficient for field trials due to variations.
7. Line 63, define “VPD”. Line 78, define the abbreviations at the first use in the main text.
8. Line 157, list the manufacturer for the CI-340 photosynthesis meter.
9. Line 178-181, please describe the statistical analysis in more details.
10. Line 222-226, please explain the calculation of the LL equations in sufficient details, as this is an important result to support the conclusion of this study.
11. Line 385, list the university of the Department of Horticulture Science.
12. In the figure legends, “It is indicates” should be removed. The author should rewrite the figure legends to better describe the content of the figures.
13. None of the figures had error bars. It seems like the experiment did not have replicates.
14. Please make sure the citations are in the consistent format, including consistent journal names (full name or abbreviation).
15. Table 1, it should be “Monthly weather data”, not “Weather monthly data”. The unit for temperature is “°C”, not “C°”. The author also needs to describe how was the weather data collected and recorded.

Experimental design

All my comments were listed above.

Validity of the findings

All my comments were listed above.

---

## Round 0.2 · Major Revisions

The revised manuscript has been in many aspects improved. However, several issues remain to be addressed, with most important the two issues that were arisen from me and the 1st round reviewers: Statistics and Abstract.

Statistics
In L191-196, you only mention analysis of variance. Nevertheless, the results of various statistical tests are present in the graphs/tables: Two-way ANOVA (Table 2) and regression (Figure 1), and correlation (Figure 4, although you present R2) and Multivariate ANOVA (Table 3). All the above tests should be clearly mentioned in the Statistics paragraph as well as any other relevant detail.
Notably, in Table 4 a correlation is probably presented. While you refer to correlation value in the caption of the figure, you present r2 which is the coefficient of determination (regression), although you do not present any equation of the regression. Please clarify all the above in the Statistics (Materials and Methods) and correct the caption of the Figure 4 accordingly.

Abstract
L20. Remove the “photosynthetic available radiation (Par)” since it is not a photosynthetic response (see below my relevant comment).
L27 the WW and WS are not explained here. Modify the sentence accordingly.
L27. Delete the 3 replicates because it is a not an Abstract-relevant information and gives wrong impression about your actual replication.
L27-30. I don’t think that the equations should be presented in the Abstract.
L30 Tc-Ta: explain at first appearance and not in L32-33 which is the second appearance.
L33. Remove PAR and the relevant percentage.
L39 the word “compared” is confusing and possibly wrong. Please rephrase as follows: “… and photosynthetic performance under both well-watered and drought conditions”.

Other major issues
PAR (Photosynthetically Active Radiation): Although plant leaf can modify the photon flux density of PAR incident on leaf blade through changes in inclination, PAR is not an attribute of the plant and its use as “photosynthetic characteristic” throughout your MS is rather problematic. Please rephrase the relevant parts dealing with your PAR measurements as a result of leaf inclination which is in turn result of WS. Please do the modifications in all relevant parts in Abstract and in L226, L330, and L399.

Units in photosynthetic parameters: L163-164. The units are wrong. According to the relevant values you present in Table 3, the correct units are as follows; Pn in μmol m-2 sec-1, gs in mmol m-2 sec-1, E in mmol m-2 sec-1.

Figure 3 presents three issues: a) Correct the gs units as per my previous comment. b) Please complete the color legend with the 4 missing lines: gs_WW_33, gs_WS_33, gsWW-59 and gs_WS_59. c) The figure does not depict a regression or other relationship, thus the CWSI x gs in the caption is not justified. Please change it to “Hourly CWSI plotted together with gs values of genotypes under different irrigation levels”. The latter point should be taken into consideration also in the Results part in L269, and Discussion in L374, where the CWSI x gs should be modified to “combined diurnal CWSI and gs graph”.

Concerning the aims of your study: you refer, among others, to “contribute to the calibration of image-based remote sensing studies, a goal which you neither discuss in the Discussion section, nor you relate your results with. You only mention this goal in Abstract, Introduction and Conclusions, i.e. in the parts that you refer to aims, but not in the actual results of your study or any of your data. Please either put it in the context of your study’s output in your Discussion or delete it. I suggest the latter option, since the other aims you note (“to determine the most suitable variety for the region by determining the yield, photosynthetic responses, and CWSI-IWUE relationships and the irrigation program based on CWSI”) are well supported by your results and are well discussed.

L63-66. Before this part I think you should consider the important comment of Reviewer 2 (of the first cycle), which you have not addressed adequately, concerning the definition of IR in the frame of its relationship with Epan. The reviewer 2 had noted that “The author needs to explain the definition of IR in the introduction, not just in the experimental design. And the author needs to explain why is IR50 considered water stress (WS) condition for strawberries.” Here, you should incorporate the reference you used to your response to Reviewer. If you sufficiently connect IR with Epan, then you can use the various labels of IR (50-125) otherwise they cannot stand alone there.

L360-361. It is not correct to argue that leaf size causes drought tolerance. Possibly the leaf size is related to thermoregulation as your reference (Reginato, 1983) implies, but there is not a direct cause-effect relationship between leaf size and drought tolerance, with the latter resulting from a suite of physiological and biochemical and anatomical traits and processes. Please correct the sentence accordingly.

Minor issues

L47-49 and L49-52: Please exchange the position of the two sentences. After the initial sentence of growing world population, the reference to the FAO predictions should follow and then the additional problems of water scarcity etc. After that, the text about tolerant and productive strawberry genotypes continues well.
L52. Correct the author of the reference
L54. The name of the species should be mentioned in the first appearance: Fragaria chiloensis and hereinafter the F. could be used (except of the case Fragaria x ananassa).
L64. Remove “at”
L82-85: correct to: Furthermore, Li et al. (2019) has pointed out the necessity to make calibrations by determining the real CWSI values with field tests in order to obtain automatic CWSI values according to the conditions of each region.
L87-88: make clearer and more concise the part of the sentence concerning CWSI (from “no studies” to the end). What does “CWSI will be felt most in the future” mean when you are referring to an index?
L156-159. The first part of the sentence until “ten plants” needs revision to be clearer. Concerning the calculation of IWUE something’s wrong: you divide the amount of water by the ratio of marketable fruits? Which ratio? And why the units of IWUE are g/mm? Moreover, the Hao et al. (2014) you referenced to at this point calculates the WUE (Kg/m3) as “the ratio of grain yield to seasonal ET”. Please correct.
L168. Replace were with was
L169. Of 3 ???
177. replace the “as mentioned” with “according to”
L183-184. A verb is missing from the sentence and replace semicolon with comma.

Figure 2. In the legend: move the 59 below 33 and then the indications of water treatments.
The page with the caption of Table 2 contains the caption of Table 3. Please correct.

L237-238. The definition of LL and UL should be entered in the first appearence of them in L180 (Materials and Methods).
L239. Correct to “In the absence of water stres, the equations ....”
L272-273. Several grammatical errors in this sentence, please correct to “in the evening to protect plants against…”
L269. Correct 70% Etp
L312-314. Please rephrase the sentence to be clearer concerning to which parameters the values correspond to.
L332. You have not measured photosynthetic capacity but photosynthetic performance. The former needs light response curves and saturating light, because it refers to the maximum photosynthetic rates.
L354. Define or delete YSP.
L378. Researches or studies, but not research
Conclusions. Please delete the first paragraph as it is a copy of Abstract/Introduction parts.

---

## Round 0.3 · Minor Revisions

The R2 was significantly improved and most of my suggestions and corrections were incorporated. However, two major corrections are needed in Figure 4 and IWUE calculation. Additionally, some minor issues remain, mainly related to comments on the R1 version that have not been considered, as I specifically analyze below.

L20-21. In order to avoid the many parentheses that make reading more difficult, I suggest changing to “(net photosynthesis, Pn; stomatal conductance, gs; transpiration rate, E)”
L22: Please replace the “irrigation level” with “irrigation regime”, so that the abbreviation IR makes sense.
L51: F. x ananassa, please correct to Fragaria x ananassa
L157-158. Two issues here: the units and the correct division. You write “IWUE (g mm-1) was determined by dividing the total amount of irrigation water used by the marketable fruit yields”. However, the correct is rather the opposite: fruit yields (in grams) divided by the total amount of water (in m3 or other volume unit). As I mentioned in my R1 comments “the Hao et al. (2014) you referenced to at this point calculates the WUE (Kg/m3) as “the ratio of grain yield to seasonal ET””. Please correct both the text and the units of IWUE throughout the MS and the relevant figure.
L228. According to my comments on R1, you do not measure photosynthetic capacity but photosynthetic performance. So please, correct to “…photosynthetic performance, which may be partly ascribed to lower PAR incident in leaf blade”
L275. You have not corrected my R1 comment “Several grammatical errors in this sentence, please correct to “in the evening to protect plants against…”
L355. You have replaced YSP with LWP, but there is no explanation of this abbreviation in your MS. Please write it in words and not as abbreviation.

L361-363. Please re-visit these two sentences because they are highly problematic: the first one (L361-362) makes no sense, especially the “ability of thermoregulation of genetics”. Please change it. Also, the Figure 5 has no apparent connection with the text in this sentence.
The second one (L362-363) could be terminated at “by temperature”, unless the referred article clearly demonstrates that small sized leaves have lower thermoregulation efficiency.

Figure 3. You have not changed the caption according to my comment in R1: “Please change it to “Hourly CWSI plotted together with gs values of genotypes under different irrigation levels”.
Figure 4. You present two regression lines with only one equation. The equation is obviously wrong since you have negative relationship, and additionally the numbers make no sense. Please present for each regression line the: a) R2, b) p value, c) correct equation.
Figure 4 caption. In the figure you use abbreviations of the genotypes, which should be explained in the caption.
Figure 5. In the caption should be explained which photo corresponds to each genotype.
Finally, another comment on the R1 that was not considered: “The page with the caption of Table 2 contains the caption of Table 3. Please correct.”

---

## Round 0.4 · Minor Revisions

You have exchange Figure 4 with Figure 5 when building your PDF. In the main body of the manuscript it is OK, i.e. the Figure 4 refer to CWSI x IWUE and the Figure 5 contains the photographs. However, when you uploaded the two figures to the Peerj system, the numbering of the figures was wrong. Please correct.

L40. No keywords have been added.

L363. Replace to “…better ability to regulate leaf temperature under…”

Figure 4. You have used different captions in the main manuscript text and in the end of the revised PDF. Please use the first one since is more correct and representative of your CWSI x IWUE figure.

Concerning the caption of Table 2 that I tried to explain in the R2 and your response (see below):

“Finally, another comment on the R1 that was not considered: “The page with the caption of Table 2 contains the caption of Table 3. Please correct.”
(Sorry, I couldn't find the issue you mentioned. The captions of Table 2 and Table 3 are on different pages and under the correct captions in the manuscript I am working on)”

Please check the page 26 out of 29 of your Pdf; You have uploaded the following in the Peerj system when it asks you to separately add the caption:
“Different strawberry genotype and irrigation levels effects on photosynthetic responses during the active harvestin period”, which is indeed the caption of Table 3. Please correct.

---

## Round 0.5 · accepted · Accept

All the comments and corrections have been addressed. Your manuscript is ready for publication.